# Comparison of Concrete Creep in Compression, Tension, and Bending under Drying Condition

**DOI:** 10.3390/ma12203357

**Published:** 2019-10-15

**Authors:** Seung-Gyu Kim, Yeong-Seong Park, Yong-Hak Lee

**Affiliations:** Depertment of Civil and Environmental Engineering, Konkuk University, 120 Neungdong-ro Gwangjin-gu, Seoul 05029, Korea; sgdandy@konkuk.ac.kr (S.-G.K.); parkys092@naver.com (Y.-S.P.)

**Keywords:** concrete, shrinkage, tension creep, beam creep, age-dependent concrete test

## Abstract

Three types of creep experiments of compression, tension, and bending were implemented to identify quantitative relations among the three types of creep under drying atmospheric conditions. In case of the bending creep experiment, two types of unreinforced concrete beams with similar dimensions were cast for use in the beam creep and shrinkage tests. The variations in the shrinkage strain within the beam depth were measured to evaluate the effect of the shrinkage variations on the bending creep strain. The beam creep strain measured within the beam depth was composed of uniform and skewed parts. The skewed parts of the creep strain were found to be dominant whereas the uniform parts were small enough to be neglected in the bending creep evaluation. This indicated that the compressive bending creep at the top surface was close to the tensile bending creep at the bottom surface. The ratios of tensile and bending creep strains to compressive creep strain were approximately 2.9 and 2.3, respectively, and the ratio of bending creep strain to tensile creep strain was approximately 0.8. Particular attention is laid on the close agreement between tensile and compressive bending creep strains even if the creep in tension is 2.9 times larger than the creep strain in compression.

## 1. Introduction

When concrete is subjected to sustained loads in either compression or tension, creep develops, the developing mechanism is generally characterized as a “delayed” phenomenon in the category of viscoelasticity. The “delayed” phenomenon signifies a gradual progress with time immediately after a load-induced strain. Creep occurs simultaneously with shrinkage; whose increasing rate decreases as the age increases throughout the lifetime even if the Pickett effect (apparent increase in creep due to drying under loading) is not considered in a loaded specimen in hygral equilibrium with the ambient medium [1,2,3,4,5]. The total amount of strain, including the load-induced strain in concrete becomes significant while determining whether the concrete reaches the failure state, and it becomes much more critical when concrete undergoes creep in tension (tensile creep) with shrinkage because of the low strain-carrying capacity of concrete in tension [6,7,8,9,10,11,12]. Another type of creep, bending creep, has a crucial effect on the gradual increase of deflection of concrete flexural members such as the beam and slab although the question of whether the creep phenomenon is the combination of compressive and tensile creep or the independent type of creep is left behind. This paper presents the experimental results regarding to three types of creep, compressive, tensile, and bending creep.

The difference between tensile and compressive creep under drying conditions is that compressive creep is apparently small while tensile creep is significant for a fully dried concrete [1,10]. This indicates that the two delayed strain phenomenon result from different mechanisms. A few theories to address the mechanisms have been presented including microcrack theory [9,10,12,13], seepage theory [14,15,16], and viscous shear theory [14,15,17]. Numerous experiments have been carried out to figure out the internal developing mechanisms of tensile and compressive creep under basic creep conditions, where moisture movement from or to the test specimens is prevented by sealing test specimens [9,10,11,12], to minimize the influence of shrinkage on creep that is known as the Pickett effect. Brooks and Neville [15] addressed that creep in tension is equal to or greater than creep in compression under drying conditions, however with sealed concrete little difference is observed between creep in tension and compression. Furthermore, for older concrete stored in a drying environment tensile creep may become less than compressive creep as observed in the early experiment of Illston [18]. Recent studies [9,10,12,13] also showed a similar observation as [15] even if it showed a large difference between the basic creep in tension and compression where the compressive basic creep is two to six times larger than tensile creep. Similar experimental results were also observed from many experimental studies [7,19,20,21,22]. Based on the observation that creep develops with shrinkage, whose mechanism is closely related to microcrack development in the internal microstructure, which is essentially accompanied by water evaporation to ambient medium, local stress in microstructure, and applied load [23,24,25,26]. The argument is somewhat aligned with the recognition of Ranaivomanana et al. [10] that the larger basic creep strain in compression is due to microcracks, which can be developed with shrinkage and under low level of stress in concrete [23,24]. The Pickett effect in the interaction between creep and shrinkage in compression may explain the creep development due to a microcrack [1,2,3,4,5].

The cross-section of concrete beam under a sustained load undergoes bending creep, which depends on the bending-induced normal stress states. A fundamental question regarding bending creep is whether bending creep is a manifestation of tensile and compressive creep or whether it is a tertiary type of creep mechanism. Bending creep tests have been performed with creep tests in tension and compression under basic creep conditions to probe the interrelations among the three types of creep of compression, tension, and bending [7,8,9,10,12,13,21]. A quantitative assessment of bending creep under drying conditions is hampered by the lack of available experimental results. The quantitative assessment of bending creep is narrowly available on the basis of experimental research studies under basic conditions similarly to tensile creep case because of the lack of available experimental results. Bending creep strain under basic conditions is normally smaller than the compressive creep strain and larger than the tensile creep strain [7,10,12], while it is smaller than the tensile strain and larger than the compressive strain based on few experimental results under the drying conditions [7,15]. Another question regarding bending creep is that the compressive bending creep is in close agreement with the tensile bending creep regardless of the basic or drying conditions even if the compressive creep strain is not in agreement with the tensile creep strain for both conditions. The question calls for precise measurement of the amount of bending creep that develops with shrinkage in a concrete beam. Tension-dominated creep tests are not easy to perform because cracking strain is small and much less than shrinkage strain. Further, creep strain measurements are much influenced by minute changes in the humidity level and temperature of the laboratory [7,9,12]. A recent research study [27] revealed the variation of shrinkage strain within the beam depth of a rectangular concrete beam, and interpreted the phenomenon based on two causes—the diffusion of drying, which proceeds from the surface of the concrete section, and compaction during concrete casting and bleeding [28,29,30,31,32]. The bending creep in this paper was evaluated with consideration of the variation of shrinkage within the beam depth. 

A series of bending creep tests under drying creep were implemented in the current experiment program with the compressive and tensile creep tests along with shrinkage tests under drying conditions. Compression and tension creep tests were performed on cylindrical specimens. A compression creep test was implemented by conventional creep testing apparatus with an externally installed load cell to check whether the applied external pressure maintains constant value during the test period. In the tension creep test, a testing apparatus that could hold six cylindrical specimens in series was fabricated, with a lever arm to maintain a constant pressure on the specimens. The bending creep test was performed on one-meter-long unreinforced concrete beam specimens, where the test included beam shrinkage tests and beam bending creep tests. The beam shrinkage test was designed to measure the shrinkage strains developed in the beam bending creep test. Experimental observation in the current study indicated that the compressive bending creep is in a close agreement with the tensile bending strain, which in turn, indicated that the neutral axis remained nearly unchanged during the age-dependent bending process. The compressive and tensile creep strains were measured from the cylindrical specimens in the same test environment as one used for the bending creep test and were compared to the bending creep strains to assess whether the beam creep could be presented with compressive creep. To this end, the quantitative relations among the three types of creep—compressive, tensile, and bending creep—were examined macroscopically, based on the experimental measurements of the bending, compressive, and tensile creep tests.

## 2. Test Setup

### 2.1. Test Plan

Three series of time-dependent laboratory experiments, Test Program TP-1, TP-2, and TP-3, were performed sequentially on unreinforced beam specimens and cylindrical specimens. Most creep tests have been performed to reveal the influence of different parameters on creep strain: age at loading, applied stress level, mix proportion, humidity level, and paste content on creep strain [7,9,12,15,20,21]. On the other hand, the current test program was motivated to derive a simple relation between bending creep and compressive creep strain for the purpose of applying to long-term analysis of concrete structures with normal concrete strength. The three types of creep tests were designed to identify the quantitative relations among the compressive and tensile creep strains and the bending creep strain for normal strength concrete with uniaxial compressive strength between 28 and 32 MPa. Similar mix proportions were used in each of the three test series, instead of different mix proportions, to collect sufficient experimental data for comparing the creep performances and deriving the relations among the different types of creep. The concrete mix proportions used in the three sets of experiments are listed in Table 1. TP-1 test series was aimed to evaluate the variation of shrinkage strain within the beam depth [27] as well as to identify the relation between bending creep and compressive creep strain. The water to cement ratio (W/C) of TP-1 test series in Table 1 was slightly higher than those of the other two test series of TP-2 and TP-3 to incorporate the former test aim. Two types of creep tests of compression and bending were implemented in TP-1 test series while three types of creep tests of compression, tension, and bending were implemented in TP-2 and TP-3 test series. All tests were performed for 85 ± 5 days, depending on the test process and laboratory occupation schedule. Therefore, it took approximately 180 days to complete one series of tests, including the time spent on preparing for the next test series. 

Since a testing method of concrete in tension is proposed by Elvery and Haroun [33], many testing methods have been devised and applied to the creep test in tension [7,9,10,12,13,15,20]. The testing methods were developed under similar requirements of constant load, uniaxial state, long-term measuring, and steady testing environment during the entire test period even if there is no standardized form among the methods. Three types of device depending on pressure-generating mechanism, hydraulic jack [12], screw bolt [7], and dead weight [9,10,13,15,20], were used to maintain constant load. Testing device for the tensile creep test in this study took a similar form as the dead weight case that equips a frame-type flail lever arm.

It is not easy to perform tension-dominated creep tests under drying conditions because the strain corresponding to the tensile strength is small (1–1.5 × 10^−4^) and the portion of creep strain is much less than that of the shrinkage strain in the measured total strain. Creep strain measurements are influenced by minute changes in the humidity level and temperature of the laboratory [15,16]. The influence of temperature can be encountered when the electric wire to connect strain gauge with data acquisition system is exposed to the temperature change of more than 2 °C because of the elongation of wire length. The three series of experiments were carried out sequentially in a controlled room at a constant temperature of 20 ± 2 °C and relative humidity of 60% ± 2%. All the specimens were covered with nonwoven fabric so that they were not exposed to the air and were fully moisturized by sprinkling water until the formworks were removed. The formworks were removed two days after concrete casting, for the three test series, and the specimens were cured under a humidity of 95%, until the day before the gauge attachment. The strain gauge with 60 mm length was attached on the surface of concrete specimen with glue and measured strains during the entire test periods for TP-1, TP-2, and TP-3. Strain gauges were connected to data acquisition system through electric wire. All the measured strain data was stored in a data acquisition system that consists of two data logger with 30 data channels per logger. A switching box with 50 data channels was connected to each data logger to extend the number of data channels.

The strain gauges were installed at the age of 8 days, 10 days, and 11 days for TP-1, TP-2, and TP-3 test series, respectively. The age-dependent responses were measured for 80, 86, and 90 days for the TP-1, TP-2, and TP-3 test series, respectively. The gauge readings from the specimens were stored automatically, every minute, in the data-acquisition system. All the stored data was extracted from the data-acquisition system at five-day intervals, to avoid accidental data loss. Elastic modulus and compressive and tensile strengths of test concrete for TP-1, TP-2, and TP-3 were measured from the three cylindrical specimens where four strain gauges were attached at the mid-height of a cylindrical concrete specimen (150 mm diameter and 300 mm height) at angles of 90°. Table 2 summarizes the averaging material properties obtained from three cylindrical specimens at the age of loading and 28 days for TP-1, TP-2, and TP-3 test series. Tensile strength in Table 2 is the splitting tensile strength that was obtained by tensile splitting test (or indirect tension test) where Brooks and Neville [15] presented that splitting tensile strength is close to a tensile strength under 3 MP of tensile strength.

### 2.2. Creep Tests Under Compression and Tension

Cylindrical specimens with a diameter of 150 mm and height of 300 mm were used in the compressive creep test (CR-C) and the tensile creep test (CR-T). The shrinkage strains of CR-C and CR-T were measured from similar cylindrical specimens and were designated as SH-C and SH-T, respectively. Two conventional creep-testing apparatuses were used to measure the compressive creep strains under a constant pressure of 5 MPa for TP-1 and TP-2 and 6 MPa for TP-3, where each testing apparatus mounted two cylindrical specimens vertically (Figure 1a). The stress ratios to strengths were 24.5%, 21.6%, and 25% for TP-1, TP-2, and TP-3, respectively. The stress levels in the current test programs were below 30–50% stress level, which indicates the development of linear creep development [1,7,9,12,13,15,20]. Hydraulic pressure was applied by a hydraulic jack. The pressure level was monitored by a pressure meter attached to the jack and a load cell mounted at the top of the specimen, throughout the testing period. The locking bolts in Figure 1a were locked immediately after hydraulic pressure was applied to prevent undesirable deformation. However, the locking of the bolts did not interfere with the release of the spring that was induced, owing to the progressive deformation of the specimen. The loss of axial pressure due to the release of the spring was supplemented every two days to maintain the pressure level, as designated in each test series.

The test instrument for CR-T was designed to maintain a constant pressure on the test specimens and comprised of three parts: frame structure, loading system, and circular beam connector (Figure 1b). The frame structure was designed to withstand the load that was applied internally to the specimens within the testing instrument. It had three bays, with a loading system set in each bay. Two cylindrical specimens with diameter of 150 mm and height of 300 mm were vertically installed in each bay. Therefore, six cylindrical specimens could be set in the instrument during a test. A circular beam connector was designed to support three lever arms that were inserted in a circular beam with roll bearings, to eliminate friction between the circular beam and lever arms. The lever arm was fabricated to have a 1:4 distance ratio, as shown in Figure 1b. Therefore, the applied load *W* would be transferred to two cylindrical specimens in a bay, to generate a constant pressure on the specimens through the circular beam connector and lever arm. The loading system was composed of a load cell and two cylindrical specimens that were connected serially. Each constituent of the loading system was connected to a wire that was linked with a ring at both ends, to avoid any rotational resistance. One hooked end of the loading system was connected to a lever arm and the other hooked end was connected to the bottom of the frame structure to allow self-equilibrium state. Steel plates of thickness 10 mm were affixed at both ends of the cylindrical specimen, to introduce tensile pressure onto the concrete cylinder (Figure 1d). For this purpose, four steel anchors of diameter 10 mm and length 50 mm were welded on the steel plate surface, and one steel anchor of diameter 23 mm and length 60 mm was welded at the center of the steel plate surface. The outer end of the 23 mm diameter anchor was connected to a wire rope to transmit the applied load as shown in Figure 1d. The weight of 375 kgf was applied at the end of lever arm in each bay to generate 1.5 tonf of tensile load in the load cell. Steel weight that has a dimension of 200 mm × 315 mm × 50 mm and weight of 25 kgf was fabricated for the weight. Therefore, fifteen steel weights were heaped up on the loading platen of each bay in Figure 1b. The device ensured constant tensile pressure for the two cylindrical tensile test specimens during the entire test period. A constant load of 1.5 tonf induced a constant pressure of 0.8 MPa. The stress ratios to tensile strengths were 41.5% and 35.9% at the age of loading, 33% and 30% at the age of 28 days for TP-2 and TP-3, respectively. 

Strain gauges were attached at the mid-height level of the specimen, at angles of 120° to the cylindrical specimen, to measure the time-dependent development of strain during the tensile creep tests similarly to the compressive creep test. For each test series, three cylindrical specimens were placed around the tensile creep testing instrument to measure the shrinkage strains of the CR-T specimens. The same cylindrical specimens used in the CR-T test, with steel plates at both ends, were used to provide the same surface conditions as those of the CR-T test.

### 2.3. Beam Creep Test

Beam creep tests (CR-B) were performed on unreinforced rectangular beam specimens (1 m long × 150 mm high × 95 mm wide). Beam shrinkage tests (SH-B) were performed simultaneously to measure the variation of shrinkage strain within the beam depth. Therefore, the creep strain was obtained by subtracting the beam shrinkage and measured instantaneous strain from the measured total strain. The shrinkage strains of beam specimens of the three test series were measured by surface-attached 60 mm long strain gauges at three locations within the beam depth, as shown in Figure 2a,b—the top surface, mid-height surface, and bottom surface. In the case of TP-1, embedded gauges were used in addition to surface-attached gauges to probe whether variations in shrinkage were developed within the beam depth of the concrete beam, similar to the case of surface-attached gauges. The detailed descriptions for the embedded gauge installation and the test results are provided in [27].

The total strains of the three series of CR-B tests under sustained loads were measured by surface-attached strain gauges attached at the same locations as in the SH-B test cases. In addition to the strain gauges, two linear variable differential transducers (LVDTs) were installed at the bottom surface of the center span (Figure 2c), to measure the development of beam deflection with time. A concentrated load of 0.1 tonf was applied at each point 250 mm away from both the end supports, to develop a four-point bending behavior. The loads were applied to the specimens at the same loading date as that of CR-C. Total eight steel weights were heaped up on the steel plate that was supported at two loading points shown in Figure 2c. This device ensured the constant loading during the entire test period. The four-point loading induced an initial flexural normal stress of ±0.7 MPa at the bottom and top surfaces within the two loading points (Figure 2d). The stress ratios to tensile strength were 36.3% and 29.6% at the age of loading and 28 days for TP-2 test series, and 31.4% and 26% at the age of loading and 28 days for TP-3 test series, respectively. The compressive stress level was less than 4%, which is much less than the tensile stress level. 

All the specimens were cast by steel formwork. Six beam specimens were designed including three unloaded shrinkage test specimens and three loaded creep test specimens for the three test series. In TP-1 test series, two loaded creep test specimens were tested because of sudden loss of a specimen during the loading work. In the TP-2 and TP-3 test series, five and seven loaded creep test specimens were tested by adding two and four more specimens respectively, to get more experimental data for better data analysis. 

## 3. Test Results

### 3.1. Compression Creep

The age-dependent creep responses (CR-C) of the cylindrical specimens during compression were measured for four cylindrical specimens (three specimens in the case of shrinkage (SH-C)), for the three series of tests. The averages of the total strains, shrinkage strains, and creep strains for the three test series are shown in Figure 3a–c, respectively, where the average creep strain was obtained by subtracting the shrinkage strains from the total strains and averaging the results for each test series. The initial loading ages of 8, 10, and 11 days, after casting, were considered as the origins of the horizontal axes of the figures, for TP-1, TP-2, and TP-3, respectively, because of the different ages at loading for the three series of tests. The two test series of TP-2 and TP-3 showed close agreement, whereas TP-1 test series showed a relatively large difference when compared to the other tests. This was because W/C of TP-2 and TP-3 were similar, whereas W/C of TP-1 was larger than those of TP-2 and TP-3, as listed in Table 1 [34]. The specific creep strains of the three test series are presented in Figure 3d, where each specific creep strain in the figure was obtained by dividing the creep strains by the applied pressures of 5, 5, and 6 MPa for TP-1, TP-2, and TP-3, respectively. The specific creeps were used to determine the ratios of tension and bending creeps.

### 3.2. Tension Creep

Tension creep tests were performed on cylindrical specimens having the same dimensions as the CR-C test specimens, for TP-2 and TP-3 test series. Two and six specimens were used for TP-2 and TP-3, respectively. In TP-2 test series, the two specimens were mounted on the tensile creep testing instrument because of a mechanical problem in mounting more than two cylindrical specimens. After fixing the mechanical problem of the instrument (after the TP-2 series of tests), six specimens were set in three bays by mounting two specimens vertically in each bay for the TP-3 series of tests (Figure 2b). The shrinkage strains of the CR-T tests were measured from three cylindrical specimens of the SH-T tests, in addition to those of the CR-T tests, to ensure the same surface conditions as that of the CR-T specimens for shrinkage because of the steel plates affixed at both ends of the CR-T specimens. Figure 4a–c show the total strain, shrinkage strain, and tension creep strain, respectively, of TP-2 test series, where the measured total strain of the CR-T includes the instantaneous elastic strain, shrinkage strain, and tension creep strain. A tensile stress of 0.8 MPa under a tensile load of 15 kN corresponded to instantaneous strains of 35 × 10^−6^ and 36 × 10^−6^ for TP-2 and TP-3, respectively. The six tensile creep curves of Figure 4c were obtained by subtracting the averaging shrinkage strain from the six total strains measured from the two test specimens. The average shrinkage strain was calculated by averaging nine shrinkage strains measured from three SH-T test specimens. The two solid lines in Figure 4a present the averages of the total strains of the two CR-T test specimens. Four solid lines in Figure 4b present the average shrinkage strains of the three SH-T test specimens and the solid line without a legend presents the average shrinkage strain of the nine shrinkage measurements. Three solid lines in Figure 4c indicate the average tensile creep curves of the two CR-T test specimens and the solid line without a legend indicates the average tensile creep curve of the six tensile creep curves.

Figure 5a–c show the total strain, shrinkage strain, and tension creep strain, respectively, of TP-3 test series. The figures were drawn similar to Figure 4a–c. However, in the cases of (a) and (c), three strain gauge readings of a specimen were averaged, and only the resulting averaged total strain and averaged creep strain of the six specimens were shown in (a) and (c), respectively, because of the large numbers of drawings that would have resulted otherwise: eighteen measurements from six cylindrical test specimens. The solid line in (a) and (c) represents the averaged total strain curve and the averaged creep strain curve, respectively, of the six test specimens. The overall responses of the tensile creep strains of TP-2 and TP-3 test series were similar. The two averaged tensile creep strains of TP-2 and TP-3 in Figure 4c and Figure 5c show 16% difference at the age of 80 days, which is equivalent to a difference of 30 × 10^−6^, in terms of strain. The total strains of TP-2 and TP-3 in Figure 4a and Figure 5a are in close agreement, with 4% difference, which is equivalent to a difference of 17 × 10^−6^, in terms of strain. The relatively large difference between the two averaged tensile creep strains is mainly due to the difference in the two averaged shrinkage strains of TP-2 and TP-3 in Figure 4b and Figure 5b, whose difference is −2%, which is equivalent to a difference of −13 × 10^−6^ in terms of strain. Based on this observation, the 16% difference in the two averaged tensile creep strains is not significant if it is considered that the difference is almost equivalent to the difference in the two averaged shrinkage strains and that the −2% difference in the two shrinkage strains is an acceptable probabilistic variation due to the uncertainty in the nature of concrete. Therefore, an averaged tensile creep strain was computed by taking the average of the two averaged tensile creep strains of TP-2 and TP-3, and is presented in Figure 6 as a tensile creep strain curve for typical concrete with the mix proportions as listed in Table 1. Figure 6 shows that the relatively large difference between the two tensile creep strains of TP-2 and TP-3 is due to the small difference in the two shrinkage strains of TP-2 and TP-3.

### 3.3. Beam Creep

The three sets of experiments were performed sequentially in the same laboratory where the axial and tensile creep tests were performed. The number of test specimens used in the four-point bending creep tests (CR-B) was two, three, and seven for TP-1, TP-2, and TP-3 test series, respectively. Three beam specimens (SH-B) were used to measure the shrinkage strains of the CR-B tests for each test series of TP-1, TP-2, and TP-3. The shrinkage strains measured at the same locations as that used in the CR-B tests were averaged to obtain the average shrinkage strains at the top, middle, and bottom surfaces. Nine averaged shrinkage strains of three test specimens of TP-1, TP-2, and TP-3 are shown in (b) of Figure 7, Figure 8 and Figure 9, respectively; the shrinkage strains are the largest at the top, intermediate at the middle, and smallest at the bottom surfaces. The shrinkage strain variation within the beam depth was quite similar to the experimental observations of Jeong et al. [27]. They performed time-dependent beam shrinkage tests with two types of beam specimens, horizontally cast and vertically cast ones, and observed that the shrinkage variations within the beam depth was due to water bleeding and tamping during the placement of fresh concrete. The detailed research can be found in the work of Jeong et al. [27] and in the early works of Hobbs [29,30] and Hoshino [31,32].

The total strains measured at the same locations as those of the SH-B tests were averaged to obtain the averaged total strains at the top, middle, and bottom surfaces, similar to the case of shrinkage strain. Six, fifteen, and twenty-one total strains of TP-1, TP-2, and TP-3 are shown in (a) of Figure 7, Figure 8 and Figure 9, respectively. Then, the bending creep strains (c) of Figure 7, Figure 8 and Figure 9 were obtained by subtracting the shrinkage strains (b) from the total strains (a) in Figure 7, Figure 8 and Figure 9.

The bending creep strains (c) of Figure 7, Figure 8 and Figure 9 were averaged for each test series of TP-1, TP-2, and TP-3 and are plotted in Figure 10a. Figure 10a shows that the compressive normal stress above the neutral axis induced compressive bending creep strain and the tensile normal stress below the neutral axis induced tensile bending creep strain. Furthermore, the magnitudes of the bending creep strains at the top and bottom surfaces were close to each other and were consistent with small bending creep strain values at the mid-height level of the beam for the three test series. These observations are quite similar to those of [9,10,12], which were obtained from beam creep experiments under drying creep conditions. The bending creep strains at the middle of the test beams are plotted in Figure 10b, to examine their relative magnitudes when compared to the creep strains at the top and bottom surfaces. It is observed that the measured amount of creep at the middle of the specimen was comparatively lesser than that at the top and bottom, which assured that the compressive bending creep at the top was approximately equal to the tensile bending creep at the bottom and that the neutral axis under initial loading remained unchanged at the middle of the beam. 

The ratio of tensile bending creep to compressive bending creep is plotted in Figure 11a. The ratio curve of TP-2 approached 1.0 from the upper end, whereas the ratios of TP-1 and TP-3 approached 1.0 from the lower end. The ratio curve of “averaging case 1” in Figure 11a was obtained by averaging the three ratios of TP-1, TP-2, and TP-3 and that of “averaging case 2” was obtained by computing the ratio of the two bending creep strain curves at the bottom and top of the three test series (solid line curves in Figure 10a). Figure 11b shows the difference between the compressive and tensile bending creep strains, where the measured and calculated cases corresponded to the averages of cases 1 and 2, respectively, of Figure 11a. The difference was small, when compared to the amounts of bending creep strains at the top and bottom. 

## 4. Discussions

The compressive creep test is simple and reliable, when compared to the tension and bending creep tests, because of its high level of creep-inducing pressure that causes a large ratio of creep to shrinkage strain. The relations among the three creep test results were quantitatively presented for the typical concrete tested in this test program, by the ratios of tension creep and bending creep to compression creep. Figure 12 compares the averaging compressive creep strain, tensile creep strain, and compressive and tensile bending creep strains. It is noticeable that the tensile creep strain was much greater than the compressive creep strain, the compressive bending creep strain was very close to the tensile bending creep strain, and the bending creep strain was ranged between the tensile and compressive creep strains.

Figure 13 plots the ratios of tension creep to compression creep for TP-2 and TP-3 test series. Eight ratios of tension creep to compression creep were plotted in Figure 13a, where the ratios ranged between 1.6 and 3.7 and converged asymptotically as time passed. Figure 13b plots the two ratios for TP-2 and TP-3, where the two ratios and six ratios for TP-2 and TP-3 were averaged separately for each test series. In this case, the ratios converged between 3.5 (TP-2) and 2.2 (TP-3) with an average value of 2.9 (TP-2 and TP-3). Figure 14 shows the ratios of the average bending creep strain to compressive creep strain for TP-1, TP-2, and TP-3, where the average bending creep strain for each test series was obtained by averaging the two creep strains at the top and bottom, and “average” represents the mean of the three ratios for the three test series. The ratios ranged between 1.7 (TP-2) and 2.8 (TP-3). In the case of the average, it was approximately 2.3 at the age of 80 days after the initial loading. Figure 15 compares the two averaged ratios of tension creep to compression creep and bending creep to compression creep, presented in Figure 13 and Figure 14, respectively. The ratio of bending creep to tension creep was computed by dividing the two ratio curves, and was obtained as 0.8 in this test program.

## 5. Conclusions

Creep in tension and compression of concrete was studied in depth to figure out its developing mechanism in connection with shrinkage. Bending creep could be an attractive research subject for many researchers because of its developing mechanism to combine compressive creep and tensile creep form along neutral axis in concrete beam. Most researches on creep have been studied under basic conditions to reveal the fundamental creep mechanism combined with shrinkage development in micro level. The practical application to long-term analysis of concrete structures requires quantitative information for creep. The current experiment program was conducted under the research scope to identify quantitative relations among the three creep types; compressive, tensile, and bending creep under drying conditions to apply for the long-term analysis of concrete structures constructed with normal strength concrete (around 30 MPa). For this purpose, three series of creep tests were implemented with the associated shrinkage tests.

It is known that creep in tension is equal to or greater than creep in compression under drying conditions, while with sealed concrete creep in tension is equal to or less than creep in compression and bending creep is greater than creep in tension and less than creep in compression. Test results in this study identified the same observations that were used to obtain the quantitative relations among the three creep types. The ratios of the bending creep strain to compressive and tensile creep strains were 2.3 and 0.8 times, respectively. The tensile creep strain was measured from eight cylindrical specimens and compared to the compressive creep strain that yielded 2.9 ratio of tension creep to compression creep.

The bending creep test was performed on one-meter-long unreinforced concrete beam specimens. The shrinkage variations within the depth of beam was measured using the same dimensions as those of the beam specimen used in the beam creep test, to compute the bending creep strain by subtracting the shrinkage strain from the total measured strain. It was the largest at the top, intermediate at the middle, and smallest at the bottom surfaces, which was quite similar to the other experimental observations. This identified that the bending creep strain evaluation needs to account for the shrinkage variation within the beam depth that is mainly caused by bleeding and compaction of fresh concrete during casting work. The bending creep test results showed that the compressive bending creep strain was very close to the tensile bending creep strain. This indicated that the neutral axis remained unchanged during the age-dependent bending process.

The observations reported here is not conclusive for all factors, but sufficient experimental evidence was provided for the concrete within the scope of the test program. Of particular interest is that of the close agreement between compressive bending creep and tensile bending creep even if the tensile creep was almost three times larger than the compressive creep. On-going research is focusing on explaining the discrepancy by meso-level mechanics to find equilibrium state between compressive and tensile creep in beam cross-sections subjected to bending creep.

## Figures and Tables

**Figure 1 materials-12-03357-f001:**
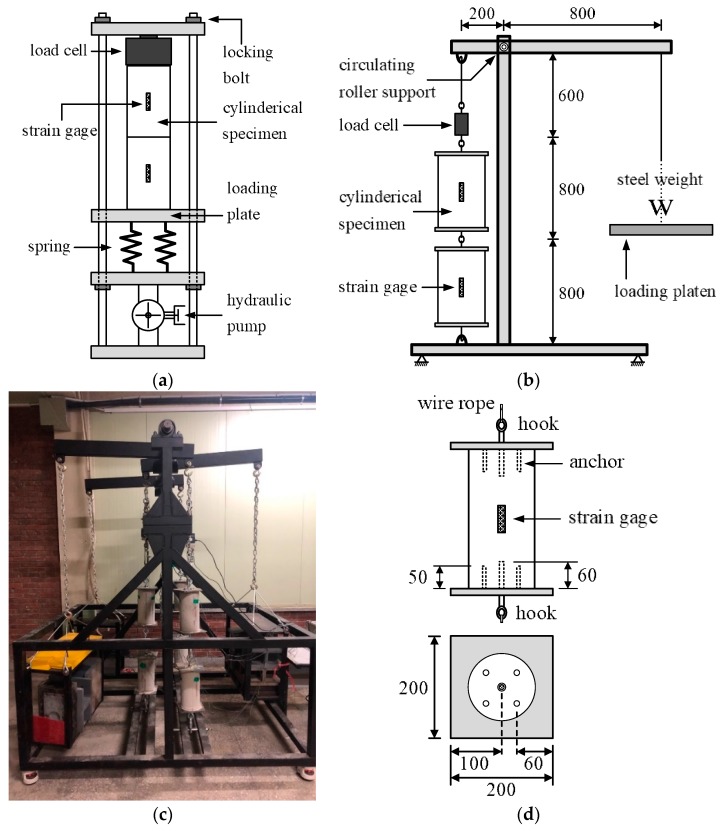
Creep test apparatus and specimen: (**a**) compressive creep testing apparatus, (**b**) tensile creep test specimen, (**c**) testing device with three sets of loading systems, and (**d**) schematic drawing of serially connected loading system (unit: mm).

**Figure 2 materials-12-03357-f002:**
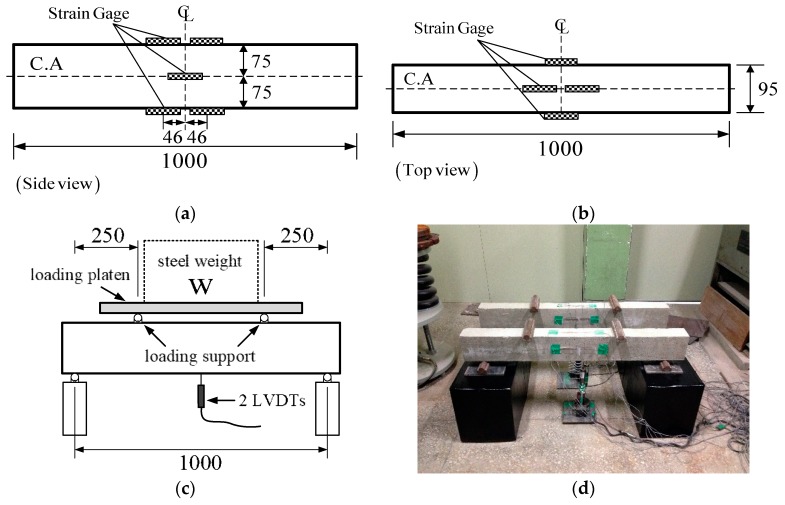
Beam creep tests: (**a**) locations of surface-attached strain gauges (side view), (**b**) locations of surface-attached strain gauges (top view), (**c**) four-point bending, and (**d**) beam creep test set-up (unit: mm).

**Figure 3 materials-12-03357-f003:**
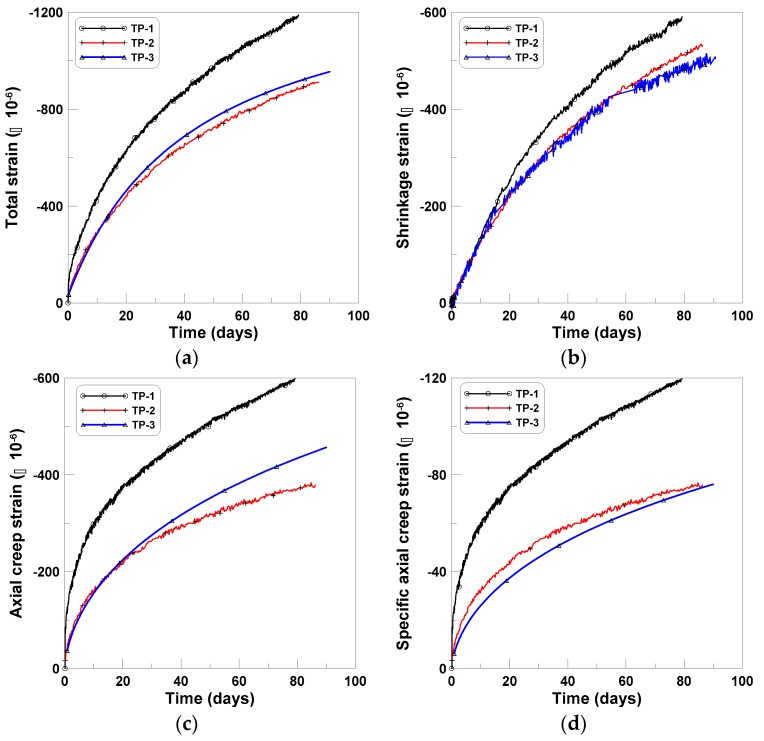
Compressive creep tests: (**a**) total strains, (**b**) shrinkage stains, (**c**) creep strains, and (**d**) specific creep strains.

**Figure 4 materials-12-03357-f004:**
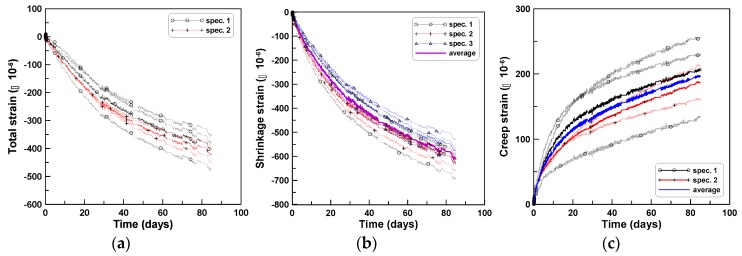
Tension creep tests of test series TP-2: (**a**) total strain, (**b**) shrinkage strain, and (**c**) creep strain.

**Figure 5 materials-12-03357-f005:**
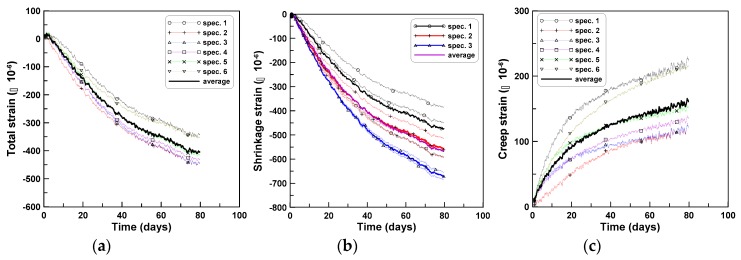
Tension creep tests of test series TP-3: (**a**) total strain, (**b**) shrinkage strain, and (**c**) creep strain.

**Figure 6 materials-12-03357-f006:**
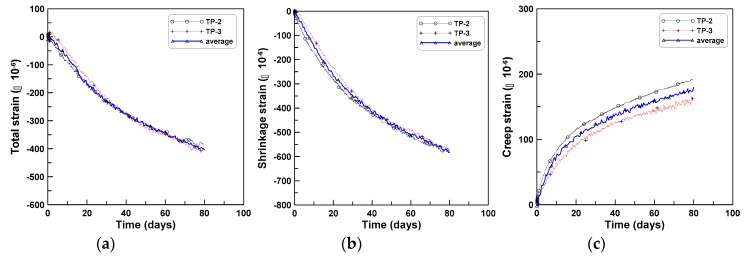
Tension creep tests of TP-2 and TP-3 test series: (**a**) total strain, (**b**) shrinkage strain, and (**c**) creep strain.

**Figure 7 materials-12-03357-f007:**
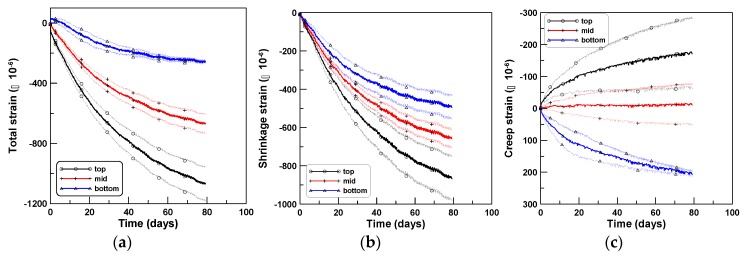
Strains at top, mid, and bottom surfaces (TP-1): (**a**) total, (**b**) shrinkage, and (**c**) creep.

**Figure 8 materials-12-03357-f008:**
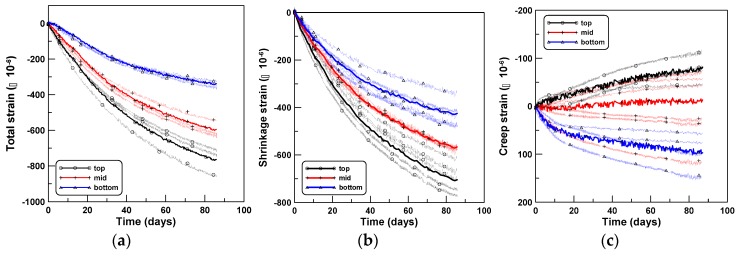
Strains at top, middle, and bottom surfaces (TP-2): (**a**) total, (**b**) shrinkage, and (**c**) creep.

**Figure 9 materials-12-03357-f009:**
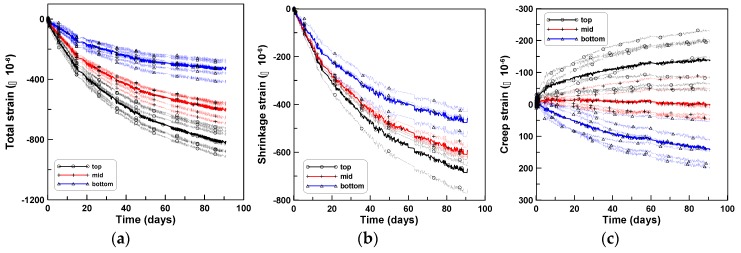
Strains at top, middle, and bottom surfaces (TP-3): (**a**) total, (**b**) shrinkage, and (**c**) creep.

**Figure 10 materials-12-03357-f010:**
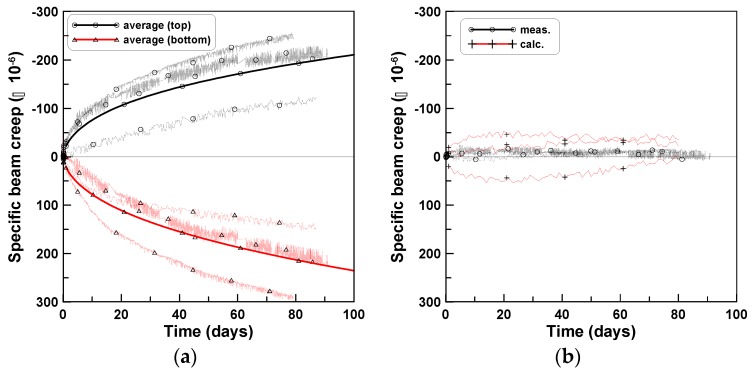
Averaging bending creep strains of TP-1, TP-2, and TP-3: (**a**) top and bottom and (**b**) middle.

**Figure 11 materials-12-03357-f011:**
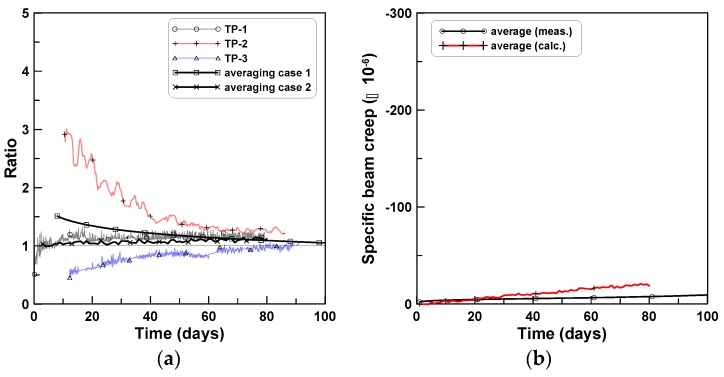
Bending creep strain ratio: (**a**) measured and calculated and (**b**) average of measured and calculated.

**Figure 12 materials-12-03357-f012:**
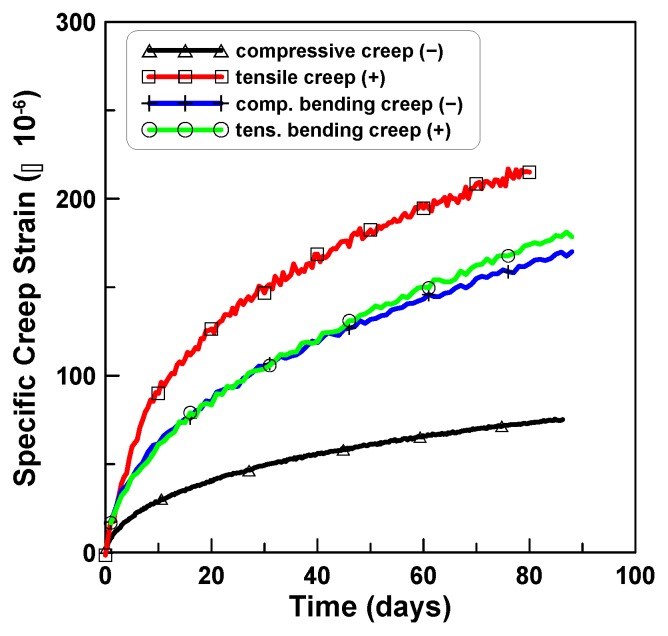
Comparison of creep in compression, tension, and bending creep.

**Figure 13 materials-12-03357-f013:**
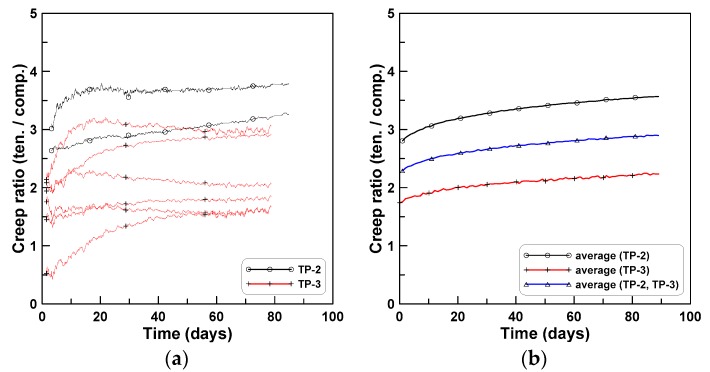
Ratios of tension creep to compression creep for TP-2 and TP-3 test series: (**a**) ratios for eight specimens of TP-2 and TP-3 and (**b**) average ratios of each test series.

**Figure 14 materials-12-03357-f014:**
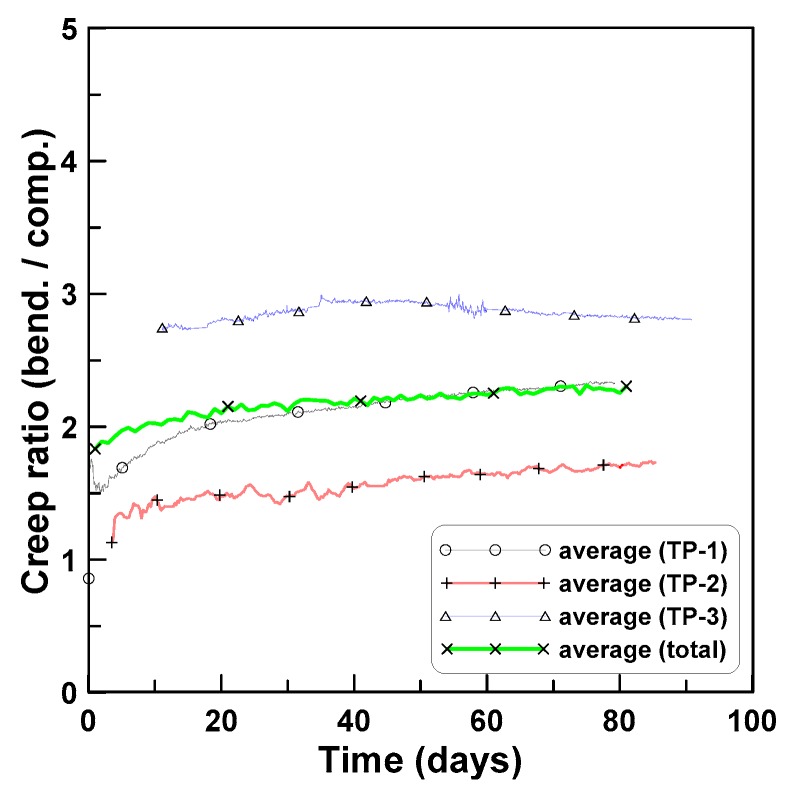
Ratios of bending creep to compression creep.

**Figure 15 materials-12-03357-f015:**
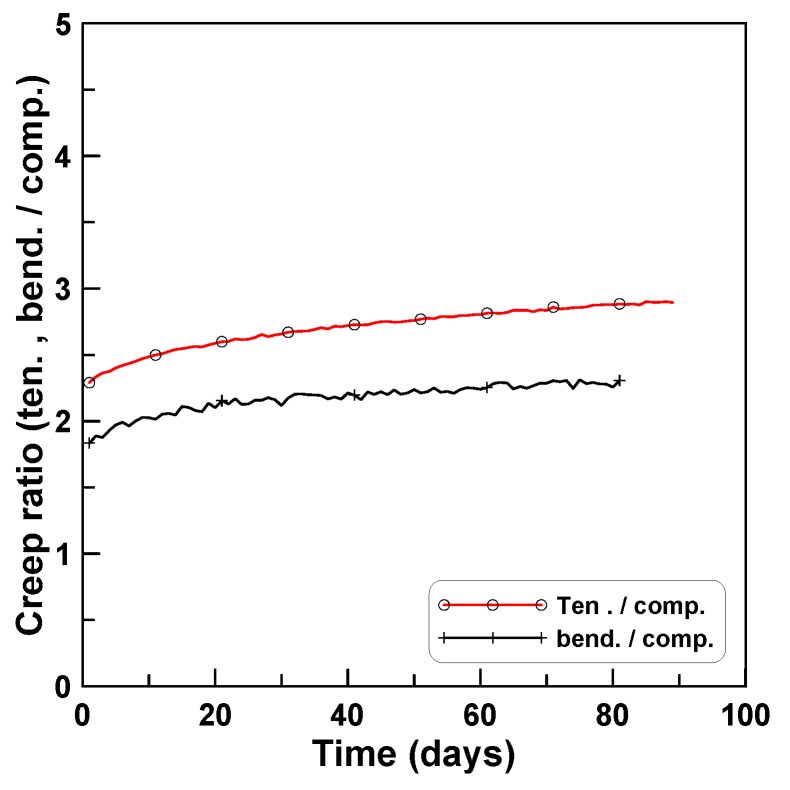
Ratios of tension and bending creeps to compression creep.

**Table 1 materials-12-03357-t001:** Concrete mix proportions for the three sets of tests.

Test Series	Max Aggregate Size (mm)	W/C Ratio (%)	Slump (mm)	Unit Mass (kg/m^3^)			
				Water	Cement	Aggregates	
						Coarse	Fine
TP-1	20	57	200	212	372	945	761
TP-2	20	54	120	190	352	974	775
TP-3	20	54	120	190	350	974	774

**Table 2 materials-12-03357-t002:** Elastic moduli and compressive strengths at 28 days.

Test Series	Age at Loading (*t_o_*, Days)	Compressive Strength (MPa)		Tensile Strength (MPa)		Elastic Modulus (MPa)	
		Age at Loading	28 Days	Age at Loading	28 Days	Age at Loading	28 Days
TP-1	8	20.4	30.0	-	-	22,700	25,300
TP-2	10	23.1	32.0	1.93	2.45	22,500	25,000
TP-3	11	24.0	33.2	2.23	2.69	22,100	25,500

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
