# Peer review of "Comparison of Concrete Creep in Compression, Tension, and Bending under Drying Condition"

_materials, 2019, doi:10.3390/ma12203357_

Round 1

Reviewer 1 Report

I enjoyed reading this experimental work. The subject is of current relevance in civil engineering and, hence, the paper falls within the scope of this journal. Nevertheless, in this reviewer's opinion, the current version of the manuscript needs MINOR REVISIONS before being reconsidered and eventually accepted for publication in this journal. 

Specific Recommendation for Revision:

Line 100: Please refer to specific standards use to conduct the test and the apparatus about compressive creep, tensile creep and beam creep.

Line 105: would the authors justify why they conduct the tension creep test only for TP-3 series?

Figure 1a: the Authors are invited to replace the figure with one better

Line 132: Please specify SH-c and SH-T

Lines 178-180:it is no clear

Lines 226-228: it is no clear

Author Response

Response to Reviewer 1

Line 100: Please refer to specific standards use to conduct the test and the apparatus about compressive creep, tensile creep and beam creep.

▶“2. Test setup” was entirely rewritten to give the descriptions in detail for testing instrument, testing method, specimen preparation, measuring devices, concrete mix proportions, data acquisition system to comply with the reviewer’s comment.

Line 105: would the authors justify why they conduct the tension creep test only for TP-3 series?

▶ Tension creep tests were conducted for test series TP-2 and TP-3. Test series TP-1 was mainly designed to investigate the shrinkage variation within the beam depth and influence of the shrinkage variation on beam bending creep. After TP-1, tension creep test was conducted to figure out the tensile bending creep occurring in bending creep. The statement is mentioned in lines 274-280 as “Tension creep tests were performed on cylindrical specimens having the same dimensions as the CR-C test specimens, for TP-2 and TP-3 test series. Two and six specimens were used for TP-2 and TP-3, respectively … ”

Figure 1a: the Authors are invited to replace the figure with one better

▶ Figure 1 and Figure 2 were merged to Figure 1 to show testing devices more clearly in the revised manuscript, and Figure 1a was removed because Figure 1b (1a in the revised) is enough to explain testing apparatus.

Line 132: Please specify SH-c and SH-T

▶ SH-C and SH-T are the shrinkage strains of CR-C and CR-T specimens. It was stated in 155-157 as “The shrinkage strains of CR-C and CR-T were measured from similar cylindrical specimens and were designated as SH-C and SH-T, respectively.”

Lines 178-180:it is no clear

▶ In TP-1 test series, two cases of strain gages, surface-attached gage and gages embedded in the inside of concrete (embedded gages), were used to measure shrinkage strains of test beam. Surface attached gages were attached on the top surface, mid surface, and bottom surface of the test beam. Embedded gages were installed above mid height, mid height and below mid height in the inside of beam specimen. The authors implemented the test program using embedded gages. However, since the shrinkage variation within the beam depth is out of scope of this study, the detailed statement regarding embedded strain gages was not included in this manuscript. The related paper was referenced in lines 217-223.

Lines 226-228: it is no clear

▶ The creep testing apparatus shown in Figure 1(b) and (c) in the revised version was composed of three bays where each bay is shown in Figure 1(b) with two cylindrical specimens vertically connected to lever arm. Figure 1(c) shows three lever arms where each lever arm belong to each bay. Statement in lines 226-228 was rewritten in lines 147-171 to include this statement.

The authors would like to express their great appreciation for the devoted comments of the reviewer that improved the quality of the manuscript.

Reviewer 2 Report

See the attached file, which is the review for the Authors.

Reviewer 3 Report

Thank you for your manuscript submission.

I have concerns with the experimental design of this research study. The number of replicates for each specimen group was uneven, and no mention of standardized tests was outlined. The study didn’t seem to have a focus when it comes to the selection of the concrete mixture designs used. What is being studied specifically from these mix designs? There was no clear variable defined (e.g. influence of water-cement ratio, influence of max aggregate size, etc.) on how it could impact the creep characteristics. In addition, no statistical analysis was conducted on the results obtained from this study.

To this end, I offer the following comments:

-Line 33 mentions the Pickett effect, but does not explain what it is until line 53. I recommend that an explanation of the Pickett effect to be given the first time it is being mentioned.

-Line 61 mentions the plane-section hypothesis, but it is not explained.

-Line 88 mentions “Experimental observations”, but does not mention by whom. Is this the authors’ previous work? If so, this should be stated (and cited if it has been published)

-In lines 92-94, why were these contradictions obtained? What caused these changes, and what was done differently?

-For Table 1, why is the mix design listed in Newtons per cubic meter? It should be listed in kg/m3. Also, why is TP-1 listed in bold font?

-What was the rationale for the mixture designs selected? They seem to be picked at random, and no mention of the influence of the water-cement ratio on shrinkage was discussed.

-Lines 119-120 mention that age-dependent responses were measured for 80,86, and 90 days for the TP-1, TP-2, TP-3 test series. Why were they all not measured for 90 days?

-Was there any reason the measuring sensors were installed at different days?

-Table 2 is confusing. I suggest adding another column to accommodate the elastic modulus at the age of loading (t=t0).

-The text makes no mention of standardized tests for measuring creep, or why it would deviate from existing standards to measure creep in the authors’ proposed method.

-The use of formulas would be beneficial to explain the relationships between total strain and creep, the specific creeps, and the creep ratios computed.

-In lines 205-208, the reason for the differences in compressive creep strains was attributed to changes in slump. This does not seem to be factual, and there is no citation to support evidence on how slump of fresh concrete properties affects creep. In addition, the difference in water-cement ratio was not discussed as a potential cause.

-Why did TP-3 have a different applied pressure than TP-1 and TP-2?

-In line 221, why did TP-2 have only two specimens, while TP-3 have six specimens for tension creep tests? Couldn’t the testing have been done after fixing the mechanical problem of the instrument?

-Why was TP-1 not tested for tension creep tests?

-Why were two, three, and seven specimens used to test beam creep for TP-1, TP-2, TP-3? Why not test all at the same number of replicas?

Author Response

Response to Reviewer 3

Line 33 mentions the Pickett effect, but does not explain what it is until line 53. I recommend that an explanation of the Pickett effect to be given the first time it is being mentioned.

▶ The statement to explain Pickett effect was given in line 29 and 45-47 as “apparent increase in creep due to drying under loading” in line 29 and “where moisture movement from or to the test specimens is prevented by sealing test specimens [9-12], to minimize the influence of shrinkage on creep that is known as the Pickett effect.” in 45-47.

Line 61 mentions the plane-section hypothesis, but it is not explained.

▶ The statement of “the plane-section hypothesis” was removed during entirely rewriting Introduction because it was replace with different expression. The statement in line 61 of the old version of manuscript was rewritten as “Experimental observations indicated that the compressive bending creep is in a close agreement with the tensile bending strain which in turn, indicated that the neutral axis remained nearly unchanged during the age-dependent bending process.” in lines 95-98 of the revised manuscript.

Line 88 mentions “Experimental observations”, but does not mention by whom. Is this the authors’ previous work? If so, this should be stated (and cited if it has been published)

▶ The statement in line 88 was modified to comply the reviewer’s comment as “Experimental observation in the current study” in line 95-96 of the revised manuscript.

In lines 92-94, why were these contradictions obtained? What caused these changes, and what was done differently?

▶ The contradictory in 92-94 of the old version of manuscript meant the close agreement between compressive bending creep at the top and tensile bending creep at the bottom even if there exists big difference between the compressive and tensile creep strain regardless of drying or basic condition. The authors recognized that the statement may be somewhat vague to lead to different interpretation. The statement was modified in lines 95-101 and mentioned in Conclusions in 445-447 as “Of particular interest is that of the close agreement between compressive bending creep at the top and tensile bending creep at the bottom even if the tensile creep is almost 3 times larger than the compressive creep.”

5. For Table 1, why is the mix design listed in Newtons per cubic meter? It should be listed in kg/m3. Also, why is TP-1 listed in bold font?

▶ Table 1 was modified to change N to kgf in line 121 as

Test

Series

Max aggregate

size (mm)

W/C ratio

(%)

Slump

(mm)

Unit weight (kgf/m3)

Water

Cement

Aggregates

Coarse

Fine

TP-1

20

57

200

2,12

3,72

9,45

7,61

TP-2

20

54

120

1,90

3,52

9,74

7,75

TP-3

20

54

120

1,90

3,50

9,74

7,74

What was the rationale for the mixture designs selected? They seem to be picked at random, and no mention of the influence of the water-cement ratio on shrinkage was discussed.

▶ The experimental program was aimed on normal strength concrete having around 30 MPa uniaxial compressive strength. For this purpose, 54-57% W/C was used to produce concrete having 28-32 MPa strength. Test series TP-1 was mainly designed to investigate the shrinkage variation within the beam depth and influence of the shrinkage variation on beam bending creep. TP-2 and TP-2 test series were focused on the quantitative relation among compressive creep, tensile creep, and bending creep. However, since TP-1 test series also included beam creep tests, beam creep test results of TP-1 was also included in the evluation of quantitative relations.

This was stated in lines 110-117 as “On the other hand, the current test program was motivated to derive a simple relation between bending creep and compressive strain for the purpose of applying to long-term analysis of concrete structures with normal concrete strength. For this purpose, three types of creep tests were designed to identify the quantitative relations among the compressive and tensile creep strains and bending creep strain for normal strength concrete with uniaxial compressive strength between 28 and 32 MPa. Similar mix proportions were used in each of the three test series, instead of different mix proportions, to collect sufficient experimental data for comparing the creep performances and deriving the relations among the different types of creep.”

Lines 119-120 mention that age-dependent responses were measured for 80,86, and 90 days for the TP-1, TP-2, TP-3 test series. Why were they all not measured for 90 days?

▶ The experiment period was designed for 90 days after introducing external load. The experiments were ended earlier than scheduled to prepare next experiments when the test data are well converged. Actually, it takes more than six months to complete one test series including casting concrete, preparing measuring devices, checking all the test instruments.

It was stated in lies 118-120 as “All tests were performed for 855 days, depending on the test process and laboratory occupation schedule. Therefore, it took approximately 180 days to complete one series of tests, including the time spent on preparing for the next test series.”

Was there any reason the measuring sensors were installed at different days?

▶ There was no special reason to install the strain gages at different days. It was because of lack of manpower for the tests. Since two research assistants had to handle the installing work, the work schedule went beyond the scheduled where the installing day was normally 8 days after concrete casting. Sometimes, it was very hard to keep the test process as scheduled.

Table 2 is confusing. I suggest adding another column to accommodate the elastic modulus at the age of loading (t=t0).

▶ Table 2 was modified to comply with the reviewer’s comment as

Test series

Age at loading

(, days)

Compressive strength

(MPa)

Tensile strength (MPa)

Elastic modulus

(MPa)

Age at loading

28 days

Age at loading

28 days

Age at loading

28 days

TP-1

8

20.4

30.0

-

-

22700

25300

TP-2

10

23.1

32.0

1.93

2.45

22500

25000

TP-3

11

24.0

33.2

2.23

2.69

22100

25500

The text makes no mention of standardized tests for measuring creep, or why it would deviate from existing standards to measure creep in the authors’ proposed method.

▶ 2. Test Setup was entirely modified to comply with the reviewer’s comment in lines 104-248. The modification was especially made in detail for specimen preparation, measuring devices, applying the load, testing method for the three creep types.

The use of formulas would be beneficial to explain the relationships between total strain and creep, the specific creeps, and the creep ratios computed.

▶ Regression analysis was not adopted, and the mean value curve was used that was obtained by averaging the raw data. The mean curves using the raw data were obtained by averaging the number of gage readings. For example, in case of beam tests of TP-2 test series having five beam creep specimens, two strain gages were attached at the top where the two gages are read with time interval because the gages were connected to different channel of data logger. In this case, total ten different gage readings (2x5) are obtained in one cycle of data reading of data logger. (It took 7 minutes to complete one cycle of reading because it takes 5 seconds for one gage reading.) Thereby, it was hard task to take an average of ten data obtained in different times. For this reason, all the data were subdivided into every six hours, and a mean value at the point of six hours was calculated by taking average for the all the data measured within six hours by dividing the number of data points. Then, mean points obtained every six hours were plotted as a mean curve. It was hard to analyze statistical confidence because of this process.

In lines 205-208, the reason for the differences in compressive creep strains was attributed to changes in slump. This does not seem to be factual, and there is no citation to support evidence on how slump of fresh concrete properties affects creep. In addition, the difference in water-cement ratio was not discussed as a potential cause.

▶ The statement in 205-208 of the old manuscript was modified in lines 260-261 as “because W/C of TP-2 and TP-3 were similar, whereas W/C of TP-1 was larger than those of TP-2 and TP-3, as listed in Table 1.”

Why did TP-3 have a different applied pressure than TP-1 and TP-2?

▶ Compressive creep tests of TP-2 and TP-3 included another test purpose of examining the effect of stepwise loading on creep responses under different initial loading. Different pressures of 5 MPa for TP-2 and 6 MPa for TP-3 were designed for it. However, the stress ratio to compressive strength was below than 30 % that is within the range of the stress ratio for linear creep development. This statement was mentioned in lines 160-162 as “The stress ratios to strengths were 24.5 %, 21.6 %, and 25 % for TP-1, TP-2, TP-3, respectively. The stress levels in the current test programs were below 30~50% stress level which indicates the development of linear creep development [1, 7, 9, 12, 13, 15, 20].”

In line 221, why did TP-2 have only two specimens, while TP-3 have six specimens for tension creep tests? Couldn’t the testing have been done after fixing the mechanical problem of the instrument?

▶ The tensile creep testing apparatus has three bays where the height of the lever arm was designed to install one specimen in TP-2 series. However, during TP-2 test, the authors decided to raise the lever arm to give enough height to install two specimens vertically in a bay. Thereby, testing apparatus for TP-2 test had three specimens capacity that adds one specimen in each bay, while that for TP-3 had six specimen capacity. One specimen was lost during the 1st installation in the bay one because of sudden release of wire connection in TP-2 test.

Why was TP-1 not tested for tension creep tests?

▶ Tension creep tests were conducted for test series TP-2 and TP-3. Test series TP-1 was mainly designed to investigate the shrinkage variation within the beam depth and influence of the shrinkage variation on beam bending creep. After TP-1, tension creep test was designed to figure out the tensile bending creep occurring in bending creep. The statement is mentioned in lines 274-280 as “Tension creep tests were performed on cylindrical specimens having the same dimensions as the CR-C test specimens, for TP-2 and TP-3 test series. Two and six specimens were used for TP-2 and TP-3, respectively .”

Why were two, five, and seven specimens used to test beam creep for TP-1, TP-2, TP-3? Why not test all at the same number of replicas?

▶ The statement to explain the reason was mentioned in lines 236-241 as “All the specimens were cast by steel formwork. Six beam specimens were designed including three unloaded shrinkage test specimens and three loaded creep test specimens for the three test series. In TP-1 test series, two loaded creep test specimens were tested because of sudden loss of a specimen during the loading work. In TP-2 and TP-3 test series, five and seven loaded creep test specimens were tested by adding two and four more specimens respectively, to get more experimental data for better data analysis.”

The authors would like to express their great appreciation for the devoted comments of the reviewer that improved the quality of the manuscript.

Round 2

Reviewer 2 Report

The revised version of the article that has been resubmitted is much better than the original submission, and has reached the form and the level to be accepted and published.

The Authors have done a good job. They have considered how I had commented their article and have suitably addressed all my comments, point by point. In so doing, they have solved all the problems.

Not only have they carefully and correctly addressed the issues raised in my review, but also they used my suggestions and criticisms to develop further personal contributions, which have increased the technical quality of the article.

Now, the article adds to the subject, which is essential for an archival scientific journal. Moreover, the presentation saves the readers’ effort to understand the article.

Author Response

The authors would like to express their great appreciation for the devoted comments of the reviewer that improved the quality of the manuscript.

Reviewer 3 Report

Thank you for the revised manuscript and responding to the reviewers' comments.

-Table 1 should not be listed as unit weight. Rather, the unit kg/m3 is referring to the mixture proportions for every cubic meter of concrete. For instance, how many kg of water is needed to make one cubic meter of concrete per the mixture design. It is a unit of mass, not force.

-It is still unclear why the testing programs had a different mixture design. It seems the purpose of the study is to derive a simple relationship between bending creep and compressive strain; wouldn't it have been easier to maintain the mixture proportions constant and test more samples? Or change one (or more) parameters to determine their effect on creep, or see if a simple relation is still applicable for a different mixture design?

-The testing period should have been kept for 90 days for each specimen for consistency. However, since the data was converging, it can be justified to some degree.

-While there was mention of how other research studies measure creep, No standardized test methods for measuring creep were mentioned in the manuscript, or how specifically this study deviates from a standardized method. This is a comment that was not addressed.

-When attributing the differences in compression creep to w/c ratios, it would be beneficial to cite work that verifies this claim.

-It is still unclear how there could be 'close agreement' between compressive bending creep at the top and tensile bending creep at the bottom, when it is mentioned that the tensile creep is almost 3 times larger? And then followed by a statement that further research is needed to explain the discrepancy? That does not mean it is in close agreement.
